# Microglia contribute to circuit defects in *Mecp2 null* mice independent of microglia-specific loss of Mecp2 expression

Dorothy P Schafer[1,2]*, Christopher T Heller[1,2], Georgia Gunner[1,2], Molly Heller[1], Christopher Gordon[1], Timothy Hammond[1], Yochai Wolf[3], Steffen Jung[3], Beth Stevens[1,4]

[1]FM Kirby Neurobiology Center, Boston Children's Hospital, Harvard Medical School, Boston, United States; [2]Department of Neurobiology, University of Massachusetts Medical School, Worcester, United States; [3]Department of Immunology, The Weizmann Institute of Science, Rehovot, Israel; [4]Stanley Center for Psychiatric Research, Broad Institute of MIT and Harvard, Cambridge, United States

**Abstract** Microglia, the resident CNS macrophages, have been implicated in the pathogenesis of Rett Syndrome (RTT), an X-linked neurodevelopmental disorder. However, the mechanism by which microglia contribute to the disorder is unclear and recent data suggest that microglia do not play a causative role. Here, we use the retinogeniculate system to determine if and how microglia contribute to pathogenesis in a RTT mouse model, the *Mecp2 null* mouse (*Mecp2[tm1.1Bird/y]*). We demonstrate that microglia contribute to pathogenesis by excessively engulfing, thereby eliminating, presynaptic inputs at end stages of disease (≥P56 *Mecp2* null mice) concomitant with synapse loss. Furthermore, loss or gain of *Mecp2* expression specifically in microglia (*Cx3cr1[CreER]*; *Mecp2[fl/y]* or *Cx3cr1[CreER]*; *Mecp2[LSL/y]*) had little effect on excessive engulfment, synapse loss, or phenotypic abnormalities. Taken together, our data suggest that microglia contribute to end stages of disease by dismantling neural circuits rendered vulnerable by loss of *Mecp2* in other CNS cell types.

*For correspondence: dorothy.schafer@umassmed.edu

**Competing interests:** The authors declare that no competing interests exist.

## Introduction

Rett Syndrome (RTT) is a devastating, X-linked neurodevelopmental disorder marked by a developmental stagnation and regression in neurological function. Early on these neurological deficits often have autistic-like features and are accompanied by an array of somatic impairments (*Chahrour and Zoghbi, 2007*; *Zoghbi, 2003*; *Lombardi et al., 2015*). Since the discovery that mutations in the transcriptional regulator Methyl-CpG-binding protein 2 (*Mecp2*) underlie the vast majority of RTT cases, studies in mouse models of RTT have implicated virtually every resident brain cell type (neurons and glia) in the disorder (*Amir et al., 1999*; *Guy et al., 2011*; *McGann et al., 2012*; *Lyst and Bird, 2015*; *Li, 2012*). However, it remains unclear which cell types primarily contribute to each phenotype and how these vastly different cell types work in concert with each other to initiate and propagate the disorder.

Microglia, the brain resident myeloid-derived cell, are among the most recent cell types implicated in RTT pathogenesis (*Derecki et al., 2012*; *Cronk et al., 2015*; *Jin et al., 2015*; *Maezawa and Jin, 2010*). However, the data have been a subject of increasing controversy

**eLife digest** Rett Syndrome is a neurodevelopmental disorder with symptoms that typically begin in girls between 6 and 18 months old. Those affected developmentally stagnate and regress – during which they lose some of their previously acquired skills and develop an array of physical impairments.

Mutations in a gene called *Mecp2* on the X chromosome cause most cases of Rett Syndrome. Mice that lack the *Mecp2* gene develop symptoms similar to those seen in people with Rett Syndrome, and so such "*Mecp2* null" mice are often used to study the disorder.

Microglia, the resident immune cells of the central nervous system, have been implicated in the development of Rett Syndrome. Introducing microglia that carry the *Mecp2* gene into *Mecp2* null mice has been shown to reduce several disease-associated abnormalities. However, exactly how microglia contribute to these changes remains unknown. In addition, a more recent report failed to reproduce these findings, and instead obtained results suggesting that microglia do not affect the development of Rett syndrome.

Schafer et al. now use the mouse visual system as a model to determine if and how microglia contribute to the development of Rett Syndrome. Like many other brain regions, the developing visual system initially has a surplus of connections between neurons, or synapses, which are subsequently pruned back. Schafer et al. previously showed in the developing visual system of early postnatal (5 days after birth) control mice (who express the Mecp2 gene) that microglia contribute to this pruning by engulfing and eliminating a subset of these excessive synaptic connections. The new experiments by Schafer et al. show that another wave of microglia-mediated synaptic pruning occurs in 40-day-old juvenile control mice.

Because *Mecp2* null mice begin to display features of Rett Syndrome when they're about 40 days old, Schafer et al. tested whether the microglia of these animals inappropriately prune synaptic connections. While this process occurred normally in neonatal and juvenile *Mecp2* null mice, microglia began to excessively engulf cells in *Mecp2* null mice when they were around 56 days old. Unexpectedly, deleting or reintroducing the *Mecp2* gene solely in the microglia of these mice had little effect on pruning activity of the microglia, and failed to affect Rett-syndrome-like symptoms in the mice.

Taken together, the data presented by Schafer et al. suggest how microglia contribute to the final stages of Rett Syndrome: by dismantling circuits of neurons that are rendered vulnerable by the loss of the *Mecp2* gene in other cell types.

(*Wang et al., 2015*). The initial study by Derecki et al. transplanted wild-type (WT) bone marrow (BM) into an irradiated mouse model of RTT, *Mecp2* null mouse (*Mecp2$^{-/y}$*(*Mecp2$^{tm1.1Jae/y}$*))prior to phenotypic symptom onset (~4 weeks of age) (*Derecki et al., 2012*). When WT BM-derived microglia-like cells engrafted the CNS, many RTT-like phenotypes were arrested and lifespan was significantly increased. While data suggested that phagocytic activity of microglia may be disrupted in *Mecp2* null mice, it remained unclear precisely how microglia were contributing to the disorder. In a follow-up study, these data were replicated using a more specific, tamoxifen-inducible Cre driver on a *Mecp2* null background (*Cx3cr1$^{CreER}$; Mecp2$^{lox-stop/y}$*) (*Cronk et al., 2015*). In addition, RNAseq analysis revealed abnormalities in glucocorticoid signaling, hypoxia responses, and inflammatory responses in peripheral macrophages and resident brain microglia isolated from *Mecp2* null mice. While these data support a role for myeloid-derived MeCP2 in RTT phenotypes and pathology, another recent study demonstrated little to no effect of re-introducing MeCP2 into myeloid cells by BM chimerism in three different RTT mouse models (*Mecp2$^{tm1.1Jae/y}$*, *Mecp2$^{LucHyg/y}$* and *Mecp2$^{R168X/y}$* mice), or by genetic expression of MeCP2 in hematopoietic cells (including microglia) in a MeCP2 null background (*Vav1-Cre; Mecp2$^{LSL/Y}$*) (*Wang et al., 2015*). Thus, it remains unclear if and how microglia, specifically, contribute to pathogenesis.

Recent work in the healthy, developing CNS has demonstrated a surprising new role for microglia in synaptic circuit remodeling and maturation (*Schafer et al., 2013*; *Tremblay, 2011a*; *2011b*; *Salter and Beggs, 2014*). Among the functions at developing synapses, we recently showed

in the retinogeniculate system that microglia contribute to the process of removing excess synapses by phagocytosing less active or 'weaker' presynaptic inputs (*Schafer et al., 2012*). Importantly, disrupting microglial phagocytic activity resulted in sustained increases in synapse density and connectivity into adulthood. In the current study, we hypothesized that microglia-mediated synaptic remodeling were abnormal in mouse models of neurodevelopmental disorders associated with aberrant brain wiring and chose RTT to test this hypothesis. In many different RTT mouse models, synaptic circuit dysfunction can be detected often prior to presentation of significant phenotypic abnormalities (*Zoghbi, 2003*; *Banerjee et al., 2012*; *Dani et al., 2005*; *Dani and Nelson, 2009*; *Nguyen et al., 2012*; *Wood et al., 2009*; *Noutel et al., 2011*; *Moretti et al., 2006*; *Medrihan et al., 2008*). This includes work in the retinogeniculate system where decreases in single fiber synaptic strength are detected at early stages of disease followed by changes in structural circuits at late phenotypic stages (*Noutel et al., 2011*). In addition, studies assessing synapse density in postmortem human and mouse brain tissue have identified abnormalities, including reductions in synapse number (*Nguyen et al., 2012*; *Chapleau et al., 2009*; *Fukuda et al., 2005*; *Jiang et al., 2013*; *Stuss et al., 2012*; *Xu et al., 2014*; *Chao et al., 2007*).

Here, using the retinogeniculate system, we examined the interactions between microglia and synapses before, during, and after the onset of phenotypic regression in the *MeCP2* null mouse (*Mecp2^{tm1.1Bird/y}*) (*Guy et al., 2001*). Furthermore, we use Cre-lox technology to specifically ablate or express *Mecp2 in* microglia and determine whether these cells play a causative role in the structural and functional synaptic abnormalities. Our data demonstrate that microglia play a role in pathogenesis of synapses by excessively engulfing presynaptic inputs at end stages of disease in the visual system; however, this effect is largely secondary and independent of microglia-specific loss of Mecp2 expression.

## Results

### Microglia engulf presynaptic inputs during a newly identified wave of synaptic refinement in the healthy, late-juvenile retinogeniculate system

The retinogeniculate system, a classic model for studying multiple waves of developmental synapse refinement, is comprised of retinal ganglion cells (RGCs) residing in the retina that project to relay neurons in the lateral geniculate nucleus (LGN) of the thalamus (*Guido, 2008*; *Hong and Chen, 2011*; *Huberman, 2007*). We previously established that microglia contribute to early phase synapse refinement by engulfing, thereby eliminating, presynaptic inputs at P5 (*Schafer et al., 2012*). Synaptic engulfment was subsequently downregulated during later waves of refinement (P9-P30) (*Guido, 2008*; *Hong and Chen, 2011*; *Huberman et al., 2008*; *Torborg and Feller, 2005*). It was unknown whether microglia regulate presynaptic input density after P30. Given that Mecp2 null mice begin to phenotypically regress ≥P30 and continue regression until premature death ~P60 (*Chahrour and Zoghbi, 2007*; *Guy et al., 2011*; *Lyst and Bird, 2015*; *Guy et al., 2001*), we first needed to establish a baseline engulfment in >P30 WT mice.

Recently, a new late wave of refinement was identified between P30 and P60 in which RGC arbors decrease in size and presynaptic boutons decrease in number (*Hong et al., 2014*). We hypothesized that microglia were contributing to this late phase refinement by transiently engulfing synapses between P30 and P60. We first confirmed a reduction in retinogeniculate synapses in the late, juvenile brain of WT mice by immunolabeling P30-P60 LGN with antibodies against the RGC-specific presynaptic protein vesicular glutamate transporter 2 (VGlut2), and the postsynaptic protein Homer1 (*Figure 1—figure supplement 1*). There was a significant reduction in the density of RGC-specific synapses (VGlut2/Homer1-positive) and a reduction in VGlut2-positive terminal size between P30 and P60 (*Figure 1—figure supplement 1A–F*). This was in contrast to VGlut1-positive corticocortical synapses, which remain unchanged (*Figure 1—figure supplement 1G–I*). To determine whether microglia contribute to late phase synaptic remodeling in the late, juvenile brain and establish a baseline to assess whether these interactions are disrupted in phenotypic *Mecp2* null mice, we next used our established assay to monitor microglia-synapse interactions in the retinogeniculate system (*Schafer et al., 2012*, *2014*). One day prior to analysis, RGC inputs were labeled by injection of anterograde dye into both eyes, cholera toxin conjugated to Alexa dye 594 or 647 (CTB-594 or

CTB-647), which is resistant to lysosomal degradation. Microglia were labeled by either genetic expression of EGFP (*Cx3CR1$^{EGFP/+}$* mice) or by immunohistochemistry using an antibody specific to the microglia-marker, Iba-1. Lysosomes that are specific to and within microglia were labeled with an antibody against CD68. Similar to previously published work (*Schafer et al., 2012*), microglia preferentially engulfed RGC inputs within the LGN at P5 (*Figure 1—figure supplement 2*) compared to older ages. However, our new data revealed a second wave of engulfment that occurred in the juvenile brain specifically at P40, which is downregulated by P50 (*Figure 1A–B*) and accompanied by a transient increase in lysosomal content within microglia (*Figure 1C*). Together with recent work demonstrating decreases in RGC arbor size and bouton numbers between P30 and P60 (*Hong et al., 2014*), our work suggests that microglia contribute to this fine-scale refinement by engulfing RGC presynaptic inputs at P40.

## Microglia engulf excessive presynaptic inputs in late phenotypic *Mecp2* null mice

Phenotypic regression is evident in *Mecp2* null mice by P40. Furthermore, these abnormalities occur after the onset of electrophysiological weakening of single fiber synaptic responses in the P20-P30 *Mecp2* null retinogeniculate system (*Noutel et al., 2011*). We hypothesized that microglia-mediated engulfment of retinogeniculate inputs in P40, juvenile mice was enhanced in *Mecp2* null mice with weakened synapses.

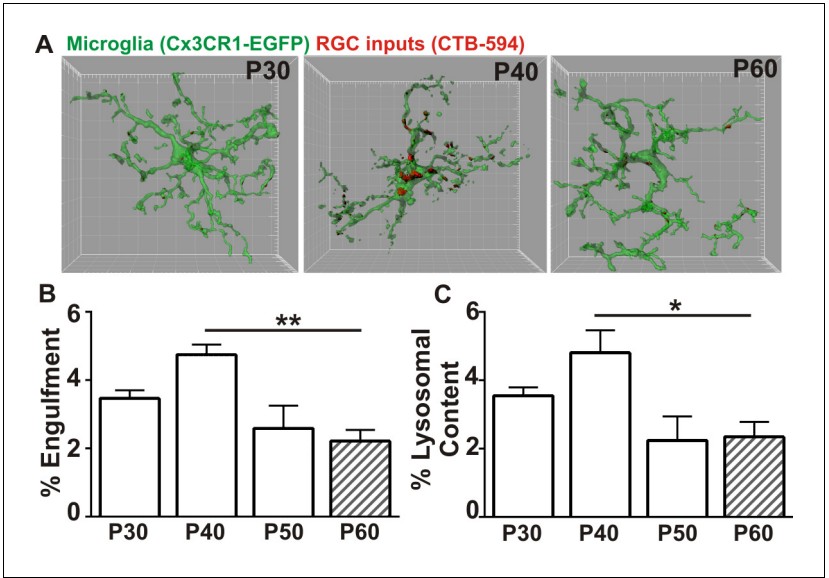

**Figure 1.** Microglia transiently engulf retinogeniculate presynaptic inputs in the juvenile P40 brain consistent with late stage synapse refinement. (**A**) Representative surface rendered microglia (green) and engulfed retinogeniculate inputs (red) from P30, P40, and P60 LGN. See also *Figure 1—figure supplement 2*. Grid line increments = 5 μm. (**B**) Quantification of engulfment reveals a transient and significant increase in engulfment of RGC inputs within microglia at P40, an age consistent with late-stage synaptic refinement (*Figure 1—figure supplement 1*). (**C**) Accompanying increased engulfment, microglia also upregulate engulfment capacity at P40 as measured by lysosomal content within each microglia (CD68 immunoreactivity per cell). *p<0.05, **p<0.01 by one-way ANOVA, Dunnett's post hoc test (all ages are compared to P60). All error bars represent SEM; N = 4–6 mice per age of mixed sex (equal ratios of males and females were used across ages).

The following figure supplements are available for figure 1:

**Figure supplement 1.** Refinement of structural synapses in the late juvenile retinogeniculate system.

**Figure supplement 2.** Presynaptic input engulfment in early and late phases of synaptic refinement in the developing retinogeniculate system.

Similar to experiments described to assess engulfment in the juvenile WT brain, RGC presynaptic inputs from both eyes were labeled with CTB-594 or CTB-647 and microglia were labeled by either genetic expression of EGFP (Cx3CR1 $^{EGFP/+}$; $Mecp2^{-/y}$ or Cx3CR1$^{EGFP/+}$; $Mecp2^{+/y}$) or immunolabeling with anti-Iba-1. In addition, to measure lysosomal content, microglia were labeled with anti-CD68. Using these methods, we detected no significant difference in microglia-mediated engulfment of retinogeniculate presynaptic inputs in P5-P50 WT or $Mecp2$ null mice compared to WT littermate controls (*Figure 2C*). However, in late phenotypic P56-P60 (≥P56) $Mecp2$ null mice, we found significant increases in engulfed inputs and lysosomal content within microglia processes and soma compared to WT littermates (*Figure 2A–D*). Consistent with engulfment being specific to synaptic compartments, we observed no significant RGC death in the retina or LGN (*Figure 2—figure supplement 1*) and we observed no changes in engulfment of other neuronal compartments including NeuN-positive somas or MAP2-positive dendrites (*Figure 2—figure supplement 2*). In addition

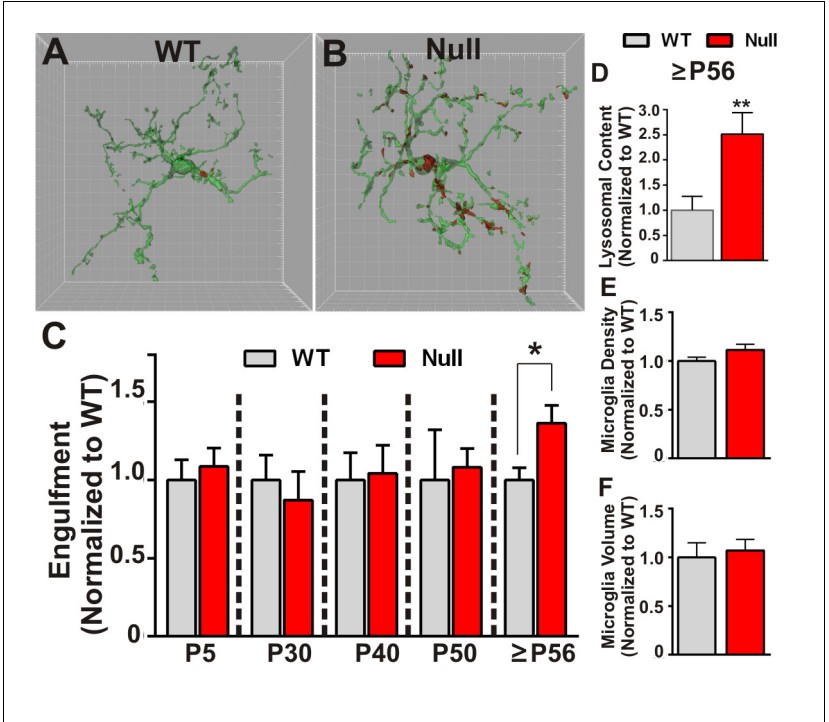

**Figure 2.** Microglia excessively engulf retinogeniculate presynaptic inputs in late phenotypic *Mecp2* null mice. (A–B) Representative surface rendered microglia (green) and engulfed RGC inputs (red) demonstrates enhanced engulfment of presynaptic inputs in ≥P56 (P56-P60) *Mecp2* null dLGN (B) as compared to WT littermate controls (A). Grid line increments = 5 μm. (C) Quantification of engulfment across development reveals excessive engulfment of presynaptic inputs within ≥P56 *Mecp2* null dLGN as compared to WT littermate controls in the absence of significant RGC cell death (*Figure 2—figure supplement 1*) or engulfment of other non-synaptic neuronal debris (*Figure 2—figure supplement 2*). *p<0.05 by multiple unpaired Student's t-tests; N = 4–6 mice per age and genotype; all data are normalized to WT controls at each age. (D) Quantification of lysosomal content (CD68 immunoreactivity) within microglia in ≥P56 LGN reveals a significant increase in phagocytic capacity in *Mecp2* null mice as compared to WT littermate controls. **p<0.01 by unpaired Student's t-tests; N = 3 mice per genotype; data are normalized to WT control. (E–F) There is no significant difference in numbers or volume of microglia within the ≥P56 LGN by unpaired Student's t-test; N = 3–5 mice per genotype; data are normalized to WT control. All error bars represent SEM.

The following figure supplements are available for figure 2:

**Figure supplement 1.** There is no significant cell death in the retinas of *Mecp2* null mice.

**Figure supplement 2.** Engulfment is specific to presynaptic inputs.

and in contrast to previously published reports (*Cronk et al., 2015*; *Jin et al., 2015*), we observed no significant changes in morphology (as measured by volume of the cell) or density of microglia, indexes of the gross, overall reactive state of these cells (*Figure 2E–F*). However, our analyses were restricted to microglia within the LGN, a region that was not analyzed previously (*Cronk et al., 2015*; *Jin et al., 2015*). These data demonstrate that, while microglia-mediated waves of synaptic engulfment are normal in the P5 and P40 *Mecp2* null brain, engulfment is excessive in ≥P56 mice— an age corresponding to late stages of phenotypic regression (*Guy et al., 2001*). Furthermore, this timing occurs after significant weakening of single fiber strength at *Mecp2* null retinogeniculate synapses (*Noutel et al., 2011*). These data suggest that microglia do not actively induce circuit defects in *Mecp2* null mice but rather facilitate late stage circuit defects by removing previously weakened structural synapses.

## Retinogeniculate presynaptic terminals and synapses are reduced in late phenotypic *Mecp2* null mice

We next assessed whether increased engulfment in P56-P60 (≥P56) *Mecp2* null mice corresponded to loss of structural retinogeniculate synapses. We first immunolabeled retinogeniculate presynaptic terminals in P40 and P56-P60 (≥P56) *Mecp2* null and WT littermate brains with an antibody directed against VGlut2. While there was no change in the density of VGlut2 immunoreactivity in P40 *Mecp2* null mice compared to WT littermate controls, there was a significant decrease at ≥P56, a time point corresponding to late-stage phenotypic regression in *Mecp2* null mice (*Figure 3A–D*). To determine whether this reduction in VGlut2 was consistent with a loss of synapses, we further assessed P56-P60 (≥P56) *Mecp2* null mice for changes in retinogeniculate synapse density defined as co-localized presynaptic VGlut2 and postsynaptic Homer1 immunoreactivity. Consistent with the reduction in VGlut2 and excessive synaptic engulfment, there was a significant decrease in retinogeniculate synapses in P56-P60 (≥P56) *Mecp2* null mice as compared to WT littermate controls (*Figure 3*) and this synapse loss was due to loss of VGlut2-positive terminals (*Figure 3C*) versus a decrease in the postsynaptic protein Homer1 (*Figure 3H*) or RGC cell death (*Figure 2—figure supplement 1*). Retinogeniculate synapses represent <10% of total synapses within the LGN (*Bickford et al., 2010*). To assess the other more abundant excitatory synapses, we immunolabeled corticogeniculate synapses with an antibody against VGlut1 within the LGN and observed no significant difference in the density of these synapses or presynaptic terminals. (*Figure 3—figure supplement 1A–E*). In addition, we assessed VGlut2 and VGlut1-positive synapse density in a neighboring thalamic nuclei (medial geniculate nucleus, MGN; *Figure 3—figure supplement 1F–G*) and observed no significant loss of these structural synapses. These results demonstrate a specific loss of retinogeniculate presynaptic terminals in late phenotypic ≥P56 *Mecp2* null mice concomitant with increased microglia-mediated engulfment of presynaptic inputs.

## Microglia-specific loss of *Mecp2* expression is insufficient to induce excessive engulfment, synapse loss, or phenotypic regression

To address how loss of *Mecp2* expression specifically affects microglia function, we crossed *Mecp2*$^{fl/y}$ mice with *Cx3cr1*$^{CreER}$ mice to conditionally ablate *Mecp2* in microglia following tamoxifen injection (*Cronk et al., 2015*; *Goldmann et al., 2013*; *Yona et al., 2013*). To achieve microglia-specific *Mecp2* ablation, tamoxifen was injected at P21-P25 and mice were assessed ~2.5 months later (P110-P120; *Figure 4A*; *Figure 4—figure supplement 1*). Consistent with microglia performing a secondary role, we found no significant increase in retinogeniculate synapse engulfment or RGC presynaptic terminal (VGlut2-positive) loss when *Mecp2* was ablated specifically in microglia (*Cx3cr1*$^{CeER/+}$;*Mecp2*$^{fl/y}$ Tam, blue hashed bars) as compared to all control groups (*Figure 4B–C*).

We also assessed other general phenotypic abnormalities known to be significantly affected in *Mecp2* null animals including overall neurological score, weight loss, rotarod performance and the optomotor task, an assessment of behavioral visual acuity previously shown to be significantly decreased in *Mecp2* null mice (*Figure 4D–G*) (*Durand et al., 2012*). We observed no significant defects in weight loss or rotarod performance in mice that lacked *Mecp2* specifically in microglia (*Cx3cr1*$^{CeER/+}$;*Mecp2*$^{fl/y}$ Tam, blue hashed bars), as compared to all controls (*Figure 4E–F*). Neurological score and optomotor task performance (*Figure 4D,G*) were also not significantly different between mice that lacked Mecp2 in microglia (*Cx3cr1*$^{CeER/+}$;*Mecp2*$^{fl/y}$ Tam, blue hashed bars) and

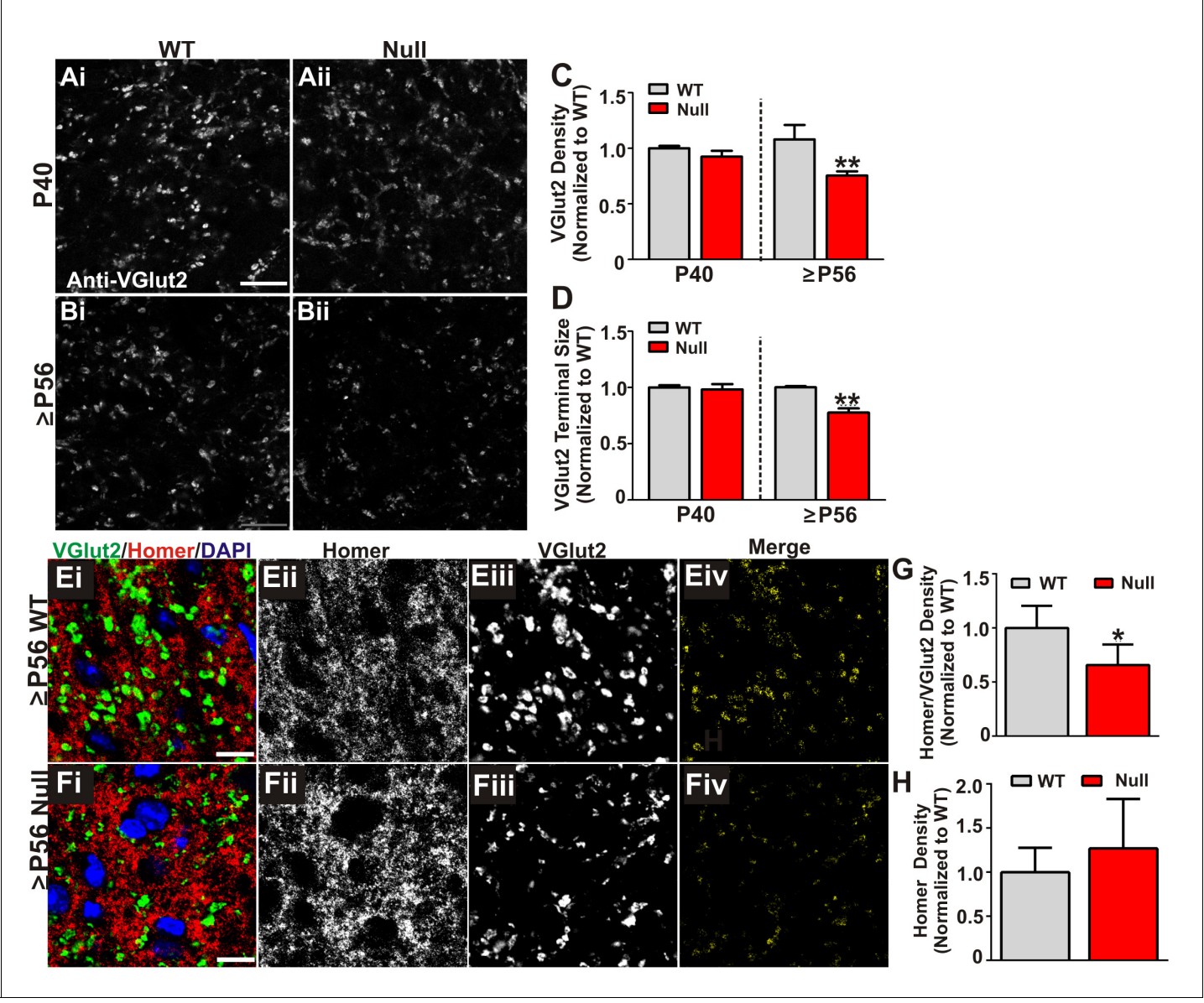

**Figure 3.** Retinogeniculate presynaptic terminals and synapses are reduced in late phenotypic ≥P56 *Mecp2* null mice. (A) Immunohistochemistry for VGlut2 to label retinogeniculate presynaptic terminals, in the dLGN of P40 (A) and ≥P56 (B, P56-P60) *Mecp2* wild-type (WT; left column) and null (right column) littermates. Images are single planes of a confocal z-stack. Scale bar = 20 µm. (C–D) Quantification of RGC presynaptic terminal (VGlut2+ puncta) immunohistochemistry reveals a significant decrease in RGC-specific terminal density (C) and size (D) in ≥P56 *Mecp2* null mice (red bars) as compared to WT littermate controls (grey bars). No significant difference was observed at P40. All data are normalized to WT control for each age. **p<0.01 unpaired Student's t-test at each age; N = 3–4 mice per age and genotype; (E–F) Immunohistochemistry for VGlut2 (green) and the postsynaptic marker Homer1 (in the dLGN of ≥P56 *Mecp2* WT (E) and null (F) littermates. Images are single planes from confocal z-stacks. The VGlut2 and Homer1 channels are separated in panels ii–iii. Panels Eiv and Fiv are colocalized VGlut2 and Homer1 puncta. Scale bar = 10 µm. (G–H) Quantification reveals a a significant decrease in RGC-specific synapses (colocalized VGlut2 and Homer) within the LGN of ≥P56 *Mecp2* null mice (red bars) as compared to WT littermate controls (grey bars) (G) and no significant change in the density of the postsynaptic protein Homer1 (H). No significant difference in density was observed in corticogeniculate-specific, VGlut1-positive synapses within the LGN or VGlut2 or VGlut1-containing synapses within a neighboring thalamice nuceli (*Figure 3—figure supplement 1*). *p<0.05 by Student's paired t-test.; N = 5 mice per genotype; all data are normalized to WT control. All error bars represent SEM.

The following figure supplement is available for figure 3:

**Figure supplement 1.** Presynaptic terminal and synapse loss are specific to retinogeniculate synapses.

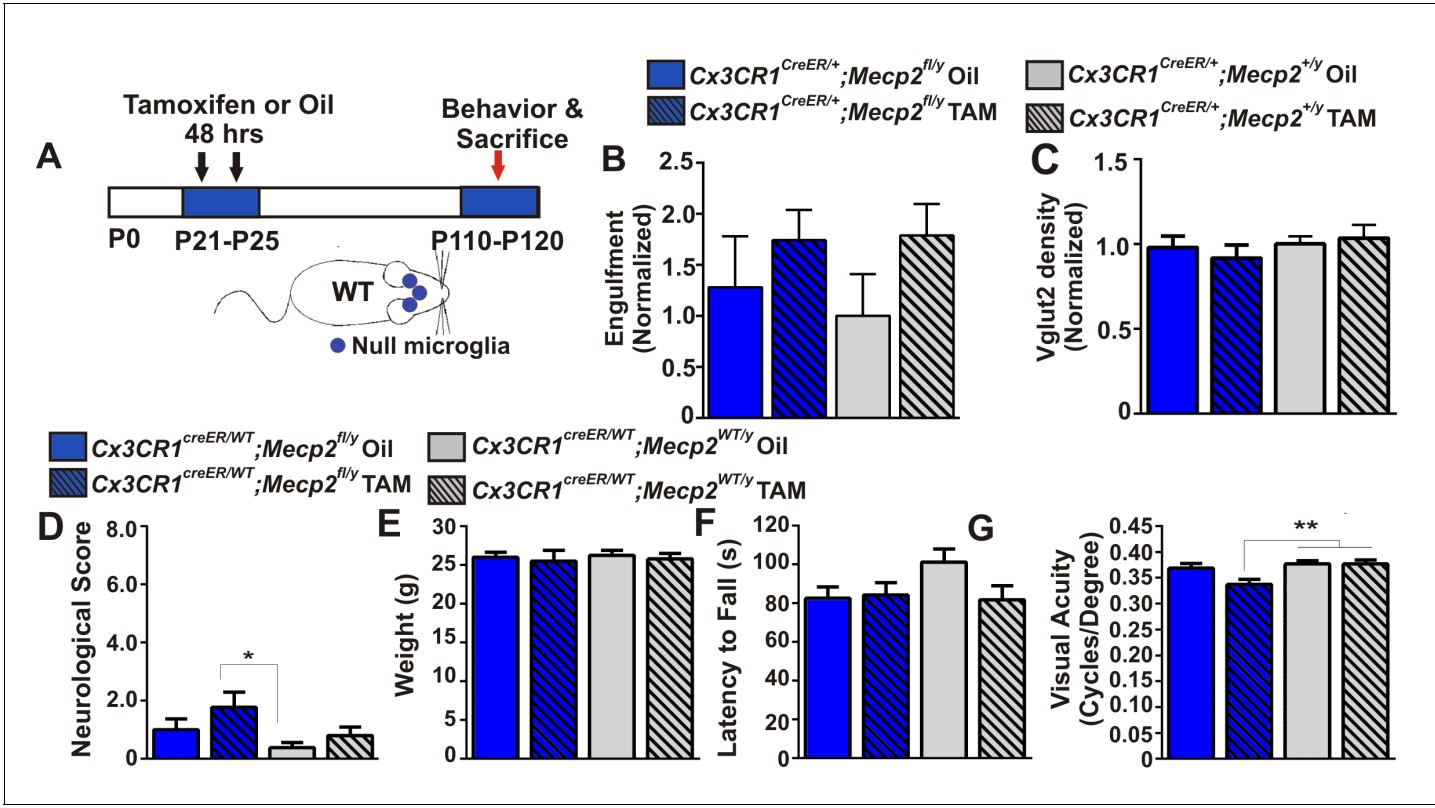

**Figure 4.** Excessive engulfment, synapse loss, and phenotypic regression are not induced following microglia-specific loss of *Mecp2* expression. (**A**) Paradigm for inducing recombination in which mice receive 2 tamoxifen or vehicle (oil) injections 48 hr apart between P21 and P25 (*Figure 4—figure supplement 1*). Behavior and postmortem analyses are subsequently performed in P110-P120 mice. (**B–C**) Quantification of engulfment (**B**) and VGlut2 terminal density (**C**) in the LGN of oil (solid bars) or tamoxifen (Tam, hashed bars)-treated mice expressing *Cx3cr1$^{CreER/+}$;Mecp2$^{fl/y}$* (blue bars) *or Cx3cr1$^{CreER/+}$;Mecp2$^{+/y}$* (grey bars) reveals no significant effect when *Mecp2* expression is specifically ablated in microglia (*Cx3cr1$^{CreER/+}$;Mecp2$^{fl/y}$* Tam, blue hashed bars) compared to all control groups. N = 4–6 mice per genotype (**D–G**). Quantification of neurological scores, weight loss, latency to fall from a rotarod, and behavioral visual acuity (optometry) in oil (solid bars) or tamoxifen (Tam, hashed bars)-treated mice expressing *Cx3cr1$^{CreER/+}$; Mecp2$^{fl/y}$* or *Cx3cr1$^{CreER/+}$;Mecp2$^{+/y}$*. There is no significant difference between mice with Mecp2-deficient microglia (*Cx3cr1$^{CreER/+}$;Mecp2$^{fl/y}$* Tam, blue hashed bars) versus the same genotype treated with oil (*Cx3cr1$^{CreER/+}$;Mecp2$^{fl/y}$* Oil blue solid bars) in any assays. However, there is a small but significant deficit in neurological score (**D**) and visual acuity (**G**) when comparing mice with Mecp2-deficient microglia (*Cx3cr1$^{CreER/+}$;Mecp2$^{fl/y}$* Tam, blue hashed bars) to WT controls (*Cx3cr1$^{CreER/+}$;Mecp2$^{+/y}$*, grey bars), an effect likely due to the hypomorphic *Mecp2$^{fl/y}$* allele. *p<0.05, **p<0.01 by one-way ANOVA, Tukey's post hoc test; N = 7–13 mice per genotype. All error bars represent SEM.

The following figure supplement is available for figure 4:

**Figure supplement 1.** Validation of Cre-mediated recombination and Mecp2 deletion in microglia.

the same genotype treated with oil (*Cx3cr1$^{CreER/+}$;Mecp2$^{fl/y}$* Oil, blue solid bars). However there was a small but significant effect when compared to WT controls (*Cx3cr1$^{CreER/+}$;Mecp2$^{+/y}$*, gray bars), an effect which may be confounded by the hypomorphic *Mecp2$^{fl/y}$* allele (see Discussion) (*Samaco et al., 2008*; *Kerr et al., 2008*). Together, these data demonstrate that loss of Mecp2 in microglia is largely insufficient to induce excessive engulfment, synapse loss or phenotypic abnormalities.

## Microglia-specific *Mecp2* expression is largely insufficient to attenuate abnormalities in microglia, synapses, or phenotypes in *Mecp2* null mice

In addition to assessing mice that specifically lack *Mecp2* expression in microglia, we did the converse experiment using a similar tamoxifen injection paradigm to express *Mecp2* specifically in microglia in an otherwise *Mecp2* null background (*Cx3cr1$^{CreER/+}$;Mecp2$^{LSL/y}$*) (*Figure 5A*, *Figure 5— figure supplement 1*). Similar mice have been assessed by other groups and have generated

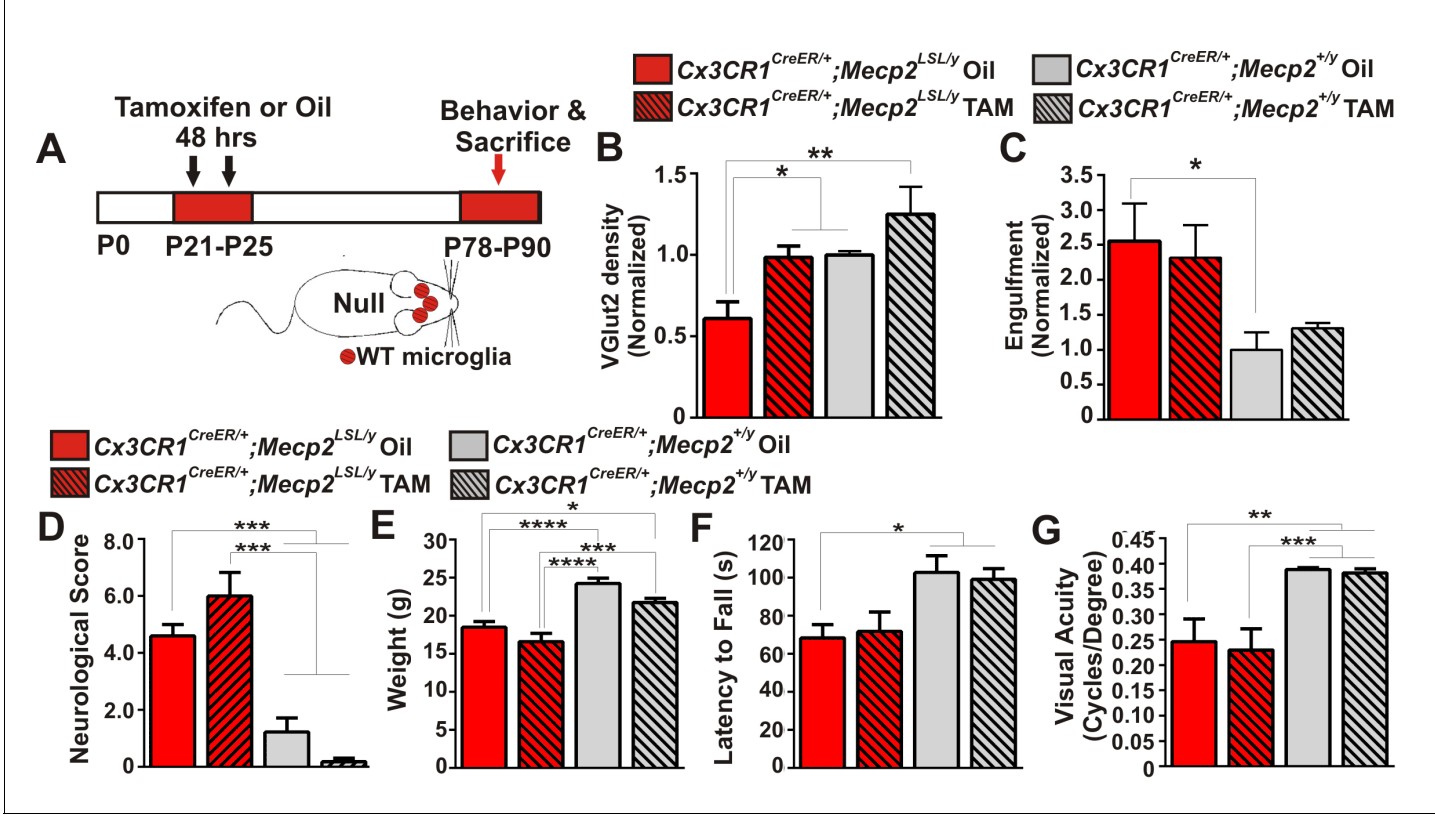

**Figure 5.** *Mecp2* expression in microglia is largely insufficient to attenuate excessive engulfment, synapse loss, or phenotypic regression in *Mecp2* null mice. (**A**) Paradigm for inducing recombination in which mice receive 2 tamoxifen or oil injections 48 hr apart between P21 and P25. Behavior and postmortem analyses are subsequently performed in P78-P90 mice (**Figure 5—figure supplement 1**). (**B–C**) Quantification of engulfment (**B**) and VGlut2 terminal density (**C**) in the LGN of oil (solid bars) or tamoxifen (hashed bars)-treated mice expressing *Cx3cr1$^{CreER/+}$;Mecp2$^{LSL/y}$* (red bars) or *Cx3cr1$^{CreER/+}$;Mecp2$^{+/y}$* (grey bars). (**B**) There was a significant decrease in VGlut2 terminal density in mice null for *Mecp2* in all cells (red solid bars) and this effect was attenuated when *Mecp2* was expressed in microglia (red hashed bars), an effect which may have resulted from tamoxifen treatment which induces a trend towards increased VGlut2 density in WT mice (grey hashed bars). *p<0.05, **p<0.01 by one-way ANOVA, Tukey's post hoc test; N = 4–6 mice per genotype. (**C**) In addition, there is a significant increase in engulfment in mice null for *Mecp2* in all cells (red solid bars) compared to WT, oil-treated littermates (grey solid bars) and this effect was not significantly attenuated when *Mecp2* was expressed in microglia (red hashed bars). However, expression of Mecp2 in microglia is no longer significant from controls, which suggests a modest effect. *p<0.05 by one-way ANOVA, Dunnett's post hoc test (all genotypes compared to *Cx3cr1$^{CreER/+}$;Mecp2$^{+/y}$* Oil, grey bars; data are not significant by Tukey's post hoc test); N = 3–5 mice per genotype. All error bars represent SEM. (**D–G**) Expression of *Mecp2* in a null background (red hashed bars) has no significant effect on attenuation of deficits in neurological score (**D**) weight loss (**E**) latency to fall from a rotarod (**F**) or visual acuity (**G**) compared to mice null for *Mecp2* in all cells (red bars). However, rotarod performance in *Cx3cr1$^{CreER/+}$;Mecp2$^{LSL/y}$* Tam mice (red hashed bar) was no longer significant from controls (grey hashed and solid bars), which suggests a modest effect. *p<0.05, **p<0.01, ***p<0.001, ****p<0.0001 by one-way ANOVA, Tukey's post hoc test; N = 6–11 mice per genotype. All error bars represent SEM.

The following figure supplement is available for figure 5:

**Figure supplement 1.** Validation of loss or gain of Mecp2 protein in dLGN microglia following Cre-mediated recombination.

differing results–one group demonstrated significant attenuation of phenotypes, while another group observed no effect (*Derecki et al., 2012*; *Cronk et al., 2015*; *Wang et al., 2015*). We sought to assess microglia dysfunction, synapse loss, and general phenotypes in *Cx3cr1$^{CreER/+}$;Mecp2$^{LSL/y}$* and clarify these disparate results.

We first assessed whether microglia-specific *Mecp2* expression was sufficient to attenuate excess engulfment and synapse loss in *Mecp2* null mice. Similar to ≥P56 *Mecp2* null mice, there was a significant increase in engulfment and decrease in RGC presynaptic terminal density in late phenotypic P78-P90 *Mecp2$^{LSL/y}$* treated with oil (*Cx3cr1$^{CreER/+}$;Mecp2$^{LSL/y}$* Oil, red solid bars) as compared to *Mecp2$^{+/y}$* controls (grey bars, **Figure 5B–C**). When *Mecp2 was expressed* in microglia in a null

background ($Cx3cr1^{CreER/+}$;$Mecp2^{LSL/y}$ Tam, red hashed bars), the excessive engulfment was not attenuated compared to the same genotype treated with oil ($Cx3cr1^{CreER/+}$;$Mecp2^{LSL/y}$ Oil, red solid bars). However, engulfment was no longer significantly different from $Mecp2^{+/y}$ controls (grey solid and hashed bars), which suggests a modest effect. In contrast, there was a significant enhancement in RGC terminal density in $Cx3cr1^{CreER/+}$;$Mecp2^{LSL/y}$ treated with tamoxifen (red hashed bars) compared to null animals (red solid bars; *Figure 5C*). Tamoxifen administration alone ($Cx3cr1^{CreER/+}$; $Mecp^{+/y}$ Tam, grey hashed bars) however, had a trend towards increased VGlut2 density in WT mice (*Figure 5C*), which suggests this effect may result from tamoxifen treatment.

We next assessed general phenotypes and behavioral visual acuity in $Cx3cr1^{CreER/+}$;$Mecp2^{LSL/y}$ mice. We did not observe a significant improvement in neurological score, weight loss, or visual acuity and only a small improvement in rotarod performance when *Mecp2* was specifically expressed in microglia (*Figure 5D–G*, $Cx3cr1^{CreER/+}$;$Mecp2^{LSL/y}$ Tam, red hashed bars). Together with data from $Cx3cr1^{CreER/+}$;$Mecp2^{fl/y}$ mice, excessive engulfment, synapse loss, and phenotypic abnormalities are largely independent of microglia-specific loss or gain of *Mecp2* expression. Our data are most consistent with recent reports that microglia-specific Mecp2 expression is insufficient to attenuate phenotypes in *Mecp2* null mice (*Wang et al., 2015*).

## Discussion

Our results demonstrate that microglia-mediated engulfment of presynaptic inputs is a plastic event that can be transiently upregulated during multiple waves of synaptic remodeling in the healthy, developing brain. Furthermore, engulfment of presynaptic inputs is upregulated and excessive in late phenotypic *Mecp2* null mice, concomitant with loss of structural synapses. Cre-lox experiments demonstrate that microglia-specific loss of *Mecp2* expression does not induce excessive engulfment or synapse loss and, similarly, gain of microglia-specific *Mecp2* expression in a null background also has little effect on attenuation of these parameters. Furthermore, deficits in general phenotypic abnormalities and behavioral visual acuity are also largely independent of microglia-specific loss or gain of *Mecp2* expression. Along with mice that express *Mecp2* specifically in microglia on a null background, these data offer significant insight into the contribution of these cells to disease progression. Taken together, our data suggest a model where loss of *Mecp2* expression in microglia has minimal effect on neural circuit integrity and function. Instead, microglia largely respond secondarily and engulf synapses in response to circuits weakened and rendered vulnerable by loss of *Mecp2* in other resident CNS cell types such as neurons and astrocytes.

### Microglia-mediated presynaptic engulfment in the healthy, juvenile brain

In the process of establishing a baseline of engulfment in the WT juvenile brain, we identified a new window of microglia-mediated presynaptic engulfment at P40 (*Figure 2*). This age corresponds to a newly identified window of late-stage, fine-scale structural synapse elimination in the retinogeniculate system (*Figure 1—figure supplement 1*) (*Hong et al., 2014*). One open question is what molecular mechanism underlies this late-stage engulfment and, if disrupted, are there sustained deficits in circuit structure and function.

In early postnatal development (first postnatal week), we previously identified that microglia engulf presynaptic inputs, in part, through complement-dependent phagocytosis (*Schafer et al., 2012*). Mice deficient in the microglial phagocytic receptor, complement receptor 3 (CR3), or complement components C3 and C1q had sustained deficits in engulfment and synaptic remodeling in the retinogeniculate system (*Schafer et al., 2012*; *Bialas and Stevens, 2013*; *Stevens et al., 2007*). Furthermore, this process was dependent upon neural activity whereby microglia preferentially engulfed less active or 'weaker' presynaptic inputs. It is unknown whether this late phase presynaptic input engulfment is dependent upon complement or activity. Given that CR3 and C3 (the ligand for CR3) decrease over development and C3, in particular, is very low/undetectable in the juvenile brain (*Schafer et al., 2012*; *Stevens et al., 2007*; *Stephan et al., 2013*), it is likely that another mechanism underlies microglia-synapse interactions in juvenile animals. It is also clear that this late-stage presynaptic input engulfment is independent of *Mecp2*, as engulfment at P40 is indistinguishable from WT littermates (*Figure 3C*). Future work to assess other molecular pathways underlying microglia-synapse interactions in the juvenile brain will be important going forward.

## Microglia excessively engulf presynaptic inputs in the *Mecp2* null brain

While in vitro work has suggested that loss of Mecp2 in microglia can affect glutamate-mediated neurotoxicity and synapses (*Jin et al., 2015*; *Maezawa and Jin, 2010*), it was unknown whether microglia affect synapses in Mecp2 null mice in vivo. Furthermore, while deficits in glutamatergic, glucocorticoid, hypoxia, and immune-related pathways have recently been reported in *Mecp2* null microglia (*Cronk et al., 2015*; *Derecki et al., 2012*; *Jin et al., 2015*; *Maezawa and Jin, 2010*), it has remained unclear precisely how microglia were contributing to disease on a mechanistic level in vivo. Our data offer significant insight into these unanswered questions. We demonstrate that microglia excessively engulf presynaptic inputs in the *Mecp2* null LGN concomitant with loss of structural retinogeniculate-specific synapses in the same region. These data are in contrast to previously published work that has suggested microglial phagocytic activity is decreased compared to WT mice (*Derecki et al., 2012*). However, this discrepancy can be explained by differences in experimental design used to measure phagocytic activity. Assays used to measure phagocytosis in previous work were in vitro in response to the addition of UV-irradiated neural precursor cells (*Derecki et al., 2012*), a context that is very different from assessing engulfment of presynaptic inputs in the retinogeniculate system in vivo. In the same study, annexin V was administered in vivo to block phagocytic activity in *Mecp2* null mice with WT BM-derived cells ($Mecp2^{LSL/y}/Lysm^{Cre}$), which resulted in failure of *WT* BM-derived cells to attenuate phenotypes in $Mecp2^{LSL/y}$ mice. However, it is unclear where and how annexin V is acting given that phagocytosis was not directly assayed in vivo and the $Lysm^{Cre}$ induces expression in many myeloid-derived cell types besides microglia. Indeed, this same group published findings that loss of *Mecp2* primarily affects peripheral myeloid-derived cell numbers and gene expression early in disease and only later affects microglia (*Cronk et al., 2015*). Going forward, it will be important to understand the contribution of these peripheral cell types to disease phenotypes.

To address whether microglia were primary or secondary to synapse loss, we used Cre-lox technology to specifically express or ablate *Mecp2* in microglia. In doing so, we demonstrate that synapse loss and excessive engulfment in *Mecp2* null mice are largely independent of microglial-specific loss of *Mecp2* expression. This is inconsistent with data in which re-expression of *Mecp2* in myeloid-derived cells (including microglia) attenuates several behavioral phenotypes and cell loss in *Mecp2* null mice (*Derecki et al., 2012*; *Cronk et al., 2015*). However, our data are consistent with data from this same group suggesting that microglia in *Mecp2* null mice are abnormal only in late phenotypic mice, suggesting a secondary effect (*Cronk et al., 2015*). Furthermore, our data are consistent with the newest report from a different group that re-expression of *Mecp2* in myeloid cells (including microglia) does not attenuate phenotypes in *Mecp2* null or mutant mice (*Wang et al., 2015*).

While our data suggest that microglia are largely secondary responders to synapses rendered vulnerable by loss of Mecp2 in other CNS cell types, the significance of excessive synaptic engulfment to disease progression is still unknown. The molecular mechanism driving secondary engulfment of synapses in *Mecp2* null mice and whether modulating this excessive engulfment results in attenuation of synapse loss are also unknown. Complement-dependent phagocytic signaling is one mechanism by which microglia have been shown to engulf synapses in the healthy brain, a pathway which is also dysregulated in disease (*Schafer et al., 2012*; *Stephan et al., 2012*; *Chung et al., 2015*; *Hong et al., 2016*; *Lui et al., 2016*). In addition, there are a number of other inflammatory genes that have been identified as dysregulated in *Mecp2* null microglia and may contribute to increased phagocytic activity (*Cronk et al., 2015*). Finally, we demonstrate that microglia are largely secondary responders in two mouse models of RTT (*Mecp2* null and $Mecp2^{LSL/y}$). It is still possible that microglia may be primary initiators of synaptic defects in other RTT models ($Mecp2^{-/+}$, *Mecp2* duplication, $Mecp2^{R270X}$ etc.), a mechanism recently reported in mouse models of frontotemporal dementia and Alzheimer's disease (*Chahrour and Zoghbi, 2007*; *Chung et al., 2015*; *Baker et al., 2013*; *Chahrour et al., 2008*; *Hong et al., 2016*; *Lui et al., 2016*). Assessing microglia function at synapses in these other disease-relevant models will be important future directions going forward.

# Minimal role for microglia in mediating phenotypic regression in *Mecp2* null mice

Ablating *Mecp2* expression specifically in microglia had little effect on phenotypic regression (*Figure 4*). The minimal effect observed when comparing *Cx3cr1*$^{CeER/+}$;*Mecp2*$^{fl/y}$ tamoxifen-treated mice to WT controls (*Cx3cr1*$^{CreER/+}$;*Mecp2*$^{+/y}$) is likely due to the *Mecp2*$^{fl/y}$ hypomorphic allele. The *Mecp2*$^{fl/y}$ mice have reduced *Mecp2* expression and develop RTT phenotypes in the absence of Cre-mediated recombination (*Samaco et al., 2008*; *Kerr et al., 2008*), effects which may become apparent if the trajectory of phenotypes were assessed after P120. Similarly, expression of *Mecp2* specifically in microglia in an otherwise null animal (*Figure 5*) had little to no effect on attenuation of any phenotype assessed. Together, our data are most consistent with the recent report that WT microglia/myeloid cells have no effect on phenotypes in *Mecp2* null or mutant mice but rather phenotypes are more likely due to loss of *Mecp2* expression in other CNS cell types such as neurons or astrocytes (*Wang et al., 2015*; *Lioy et al., 2011*; *Giacometti et al., 2007*; *Luikenhuis et al., 2004*; *Chao et al., 2010*; *Ito-Ishida et al., 2015*). Our data are in contrast to two other reports from another group that demonstrate introducing *Mecp2* in microglia and other myeloid cells on a *Mecp2 null* background results in significant attenuation of phenotypes (*Derecki et al., 2012*; *Cronk et al., 2015*). The discrepancy may result from difference in paradigms used to express *Mecp2*. For example, in our study, we induced recombination with tamoxifen in *Cx3cr1*$^{CreER/+}$;*Mecp2*$^{LSL/y}$ mice at P21-P25 and assessed phenotypes at ≥P78. This paradigm results in purely microglia-specific expression of *Mecp2* due to ongoing hematopoiesis that replaces peripheral *Mecp2*-null cells with WT cells (*Goldmann et al., 2013*; *Yona et al., 2013*). In contrast, Cronk, Derecki et al. induced recombination in these same mice at 9 weeks (~P63) *Cronk et al., 2015*. This late tamoxifen administration may be necessary to observe significant effects on phenotypes and may result from expression of *Mecp2* in peripheral myeloid cells. Furthermore, previous work by this same group demonstrated a significant attenuation of phenotypic regression in *Mecp2 null* mice after BM transplantation at P28 and engraftment with WT myeloid cells by ~P84 or with Cre mediated recombination (*Lysm*$^{Cre}$) in myeloid cells from birth *Derecki et al., 2012*. These paradigms also affect *Mecp2* expression in peripheral immune cells. Thus, we speculate that these divergent results may be due to differences in peripheral myeloid-derived cell-specific *Mecp2* expression, which is intriguing and worthy of future investigation. It should also be noted that we did not measure the entire panel of phenotypic abnormalities (breathing, open field, etc.) or survival so it is unknown if our results differ in these contexts. Finally, it is unclear how recently published data using a similar BM chimerism strategy but a different Cre mouse (Vav1-Cre) resulted in contradictory results and is also worthy of follow-up investigation (*Wang et al., 2015*).

## Summary

There have been conflicting reports regarding if and how microglia contribute to phenotypes in mouse models of RTT (*Derecki et al., 2012*; *Cronk et al., 2015*; *Wang et al., 2015*). Our data offer significant insight into how microglia contribute to disease in *Mecp2* null mice. While microglia-specific loss of *Mecp2* is largely insufficient to induce synapse loss and phenotypic regression and gain of Mecp2 in expression in *Mecp2* null mice is insufficient to attenuate these parameters, microglia contribute secondarily by dismantling synaptic circuits in complete *Mecp2* null mice. Taken together with previously published data that single fiber strength decreases during early stages of phenotypic regression in the *Mecp2* null retinogeniculate system (*Noutel et al., 2011*), we propose that microglia dismantle neural circuits in the late phenotypic *Mecp2* null brain by engulfing synapses previously rendered vulnerable and weakened by loss of *Mecp2* expression in, most likely, neurons. Given that recent studies demonstrate the reversibility of circuit defects and phenotypes in RTT mouse models (*Derecki et al., 2012*; *Lombardi et al., 2015*; *Lioy et al., 2011*; *Giacometti et al., 2007*; *Luikenhuis et al., 2004*; *Cronk et al., 2015*; *Guy et al., 2007*; *Jugloff et al., 2008*; *Castro et al., 2014*; *Garg et al., 2013*; *Patrizi et al., 2016*; *De Filippis et al., 2015*; *Ma et al., 2015*), identifying a molecular mechanism by which microglia dismantle circuits during late phenotypic stages and determining whether this is critical to end-stages of disease will be an important future directions with therapeutic potential.

# Materials and methods

## Animals

*Cx3cr1*$^{EGFP/+}$, Ai9 (RCL-tdT), *MeCP2*$^{-/y}$ (*Mecp2*$^{tm1.1Bird/y}$), *Mecp2*$^{LSL/y}$ (*Mecp2*$^{tm2Bird/y}$), and *Mecp2*$^{fl/y}$ (*Mecp*$^{tm1Bird/y}$) mice were obtained from Jackson Labs (Bar Harbor, MA) and *Cx3cr1*$^{CreER/+}$ mice were obtained from Jonathan Kipnis, University of Virginia. All mice were maintained by breeding to C57BL/6J. For some engulfment experiments, *MeCP2*$^{-/+}$ female mice were crossed with male *Cx3cr1*$^{EGFP/EGFP}$ mice. For Cre-lox experiments, *Mecp2*$^{LSL/+}$ or *Mecp2*$^{fl/+}$ female mice were crossed with male *Cx3cr1*$^{CreER/CreER}$ mice. All experiments using *Cx3cr1*$^{EGFP/+}$ or *Cx3cr1*$^{CreER/+}$ mice were performed with heterozygotes. Unless otherwise noted in figure legend, experiments were performed in male mice. For Cre-lox experiments, P21-P25 *Cx3cr1*$^{CreER/+}$-expressing mice were injected with tamoxifen (20 mg/kg; Sigma Aldrich, Natick,MA) or vehicle (corn oil; Sigma Aldrich, Natick,MA) subcutaneously two times, 48 hr apart, a protocol previously demonstrated to induce efficient recombination (*Goldmann et al., 2013*; *Yona et al., 2013*). All experiments were approved by institutional animal use and care committees and performed in accordance with all NIH guidelines for the humane treatment of animals.

## Engulfment analysis

Analysis of engulfment was performed using previously published procedures (*Schafer et al., 2012*, *2014*). Briefly, both eyes were injected with the same fluorophore-conjugated tracer (either cholera toxin β subunit conjugated to Alexa dye 594 (CTB-594) or 647 (CTB-647) (Life Technologies, Carlsbad, CA). Mice were then sacrificed 24 hr later. Brains were fixed in 4% paraformaldehyde (PFA; EMS, Hatfield, PA) for 3–4 hrs and 40 µm thick sections were prepared. Sections were further immonstained with antibodies against Iba-1 (Wako Chemicals, Richmond, VA) and/or CD68 (AbD Serotec, Raleigh, NC) to measure lysosomal content as previously described (*Schafer et al., 2012*). For analysis of non-synaptic material, adjacent brain sections were immunolabeled with antibodies against Iba-1 (Wako Chemicals, Richmond, VA), NeuN (EMD Millipore, Darmstadt, Germany), and MAP2 (EMD Millipore, Darmstadt, Germany). Sections were then imaged on a UltraView Vox spinning disk confocal microscope equipped with diode lasers (405 nm, 445 nm, 488 nm, 514 nm, 561 nm, and 640 nm) and Volocity image acquisition software (Perkin Elmer, Waltham, MA). Two LGN sections were imaged per animal and 4-63x fields of view were collected from the dorsal and ventral regions of each dLGN (8 fields of view total per animal). Images were subsequently processed in Image J (NIH) and analyzed using Imaris software (Bitplane, Zurich, Switzerland) as previously described (*Schafer et al., 2012*, *2014*).

## Synapse density quantification

Synapses were quantified similar to previously published work with modifications (*Schafer et al., 2012*). Briefly, mice were either perfused with 4% PFA followed by a 2 hr drop fix in 4% PFA or fixed identical to those methods described for engulfment analysis (see above). Tissue sections 15 or 40 µm) were subsequently prepared and immunostained for synaptic proteins. Antibodies included anti-Homer1 (Synaptic Systems GmbH, Goettingen, Germany), anti-Vesicular Glutamate Transporter 2 (VGlut2; EMD Millipore, Darmstadt, Germany), and anti-VGlut1 (EMD Millipore, Darmstadt, Germany) followed by appropriate, species-specific secondary antibodies conjugated to Alexa dyes (Life Technologies, Carlsbad, CA). Immunostained sections were imaged with a 63x Zeiss pan-Apochromat oil, 1.4 NA objective on a Zeiss LSM 700 Laser Scanning Confocal equipped with diode lasers (405, 488, 555 and 633 nm) and Zen image acquisition software (Carl Zeiss, Oberkochen, Germany). Alternatively, sections were imaged with a Leica SP8 X confocal (Wetzlar, Germany) equipped with multiple laser lines (405, 458, 488, 496, 514, and 470–670 nm white light) using a HC PL APO 63x/ 1.40 oil CS2 or a HC PL APO 40x/1.10 W motCORR (only 40 µm-thick sections) objective and LasX software. To maintain consistency across animals, the most medial dLGN sections were chosen for imaging. A total of 3 confocal z-stacks (1 µm spacing) were then collected from dorsal, medial, and ventral regions of the dLGN section. For each z-stack, (2 confocal planes with the most robust DAPI staining were subsequently chosen and analyzed for blind analysis using Image J software (NIH, Bethesda, MD). As a result, a total of 6 single confocal planes were analyzed for each animal. Fluorescent images of pre and/or postsynaptic markers were separated and thresholded blind. Density

of thresholded pre and/or postsynaptic markers were calculated using the measure particles function where a puncta size was defined and maintained for all analyses across animals for each marker (VGlut2 = 0.2-infinity; Homer1 = 0.1-infinity; VGlut1 = 0.1-infinity). The colocalization of puncta was quantified subsequently using the Image Calculator function applied to thresholded pre and post-synaptic images. The size and area of each puncta were recorded and then the total puncta area and average puncta size were calculated for each animal. The synapse or terminal densities were calculated by taking the total puncta area and dividing it by the total area of the field of view. The puncta density and puncta size were averaged across fields for each animal.

## Quantification of cell number and cell death

Samples were prepared and imaged similar to methods described for engulfment and synapse density quantification (see above). For microglia numbers, 10 x fields of vew were collected and cells were counted blind using the point tool in Image J. For cell death analysis, retinas were immunolabeled with antibodies against cleaved caspase 3 (Cell Signaling Technology, Danvers, MA), TUJ1 (BioLegend (formerly Covance) San Diego, CA) or NeuN (EMD Millipore, Darmstadt, Germany), mounted with media containing DAPI (Vectashield; Vector Labs, Burlingame, CA), and 4 fields of view (20X) were collected. Cells were counted blind for each field of view using the point tool in Image J.

## Validation of loss or gain of Mecp2 protein by immunohistochemistry

For validation of loss or gain of Mecp2 protein in microglia, a subset of tissue sections collected for engulfment or synapses analysis were selected and subjected to antigen retrieval using Retrievagen A (BD Biosciences, San Jose, CA). Briefly, sections were microwaved (power = 2) for 5 min in Retrievagen A solution. This was repeated once and then sections were washed 3 times with 0.1 M phosphate buffer. Sections were then immunostained with a rabbit antibody directed against the C-terminus of Mecp2 (a generous gift from M. Greenberg Harvard Medical School) (*Ballas et al., 2009*) and a chicken antibody against Iba-1 (Abcam, Cambridge, MA), overnight at room temperature. Sections were then washed and HRP-conjugated rabbit and Alexa fluor-conjugated rat antibodies were added to the sections for 1–2 hr at room temperature. Sections were subsequently washed and an Alexa-fluor conjugated anti-HRP antibody was added overnight at 4 degrees. After the overnight incubation, sections were mounted and imaged. It should be noted that several other Mecp2 antibodies and staining conditions were attempted, but only this antibody and condition enabled us to detect Mecp2 protein even in WT microglia. Two 63x fields of view were collected in the lateral, medial, and ventral portions of the LGN per animal (3 animals per condition) and images were assessed for Mecp2-positive microglia.

## Genomic DNA extraction for validating loss or gain of Mecp2 expression

Sorted cells were lysed and digested in TES buffer (10 mM Tris buffer, pH = 8, 5 mM EDTA, 0.1 M NaCL, 0.5% SDS and 100 ug PK) overnight in 56°C. DNA was precipitated in 70% ethanol for 30 mins at room temperature, centrifuged twice at top speed and the tubes were left to dry. Pellets were reconstituted with TE buffer for subsequent PCR. Genomic PCR for *Mecp2* gene was performed using the following primers: 5'-TGGTAAAGA CCCATGTGACCCAAG-3', 5'-GGC TTGCCACATGACAAGAC-3', 5'-TCCACCTAG CCTGCCTGTACTTTG-3'.

## Tissue extraction and flow cytometry

Brain samples were harvested from individual mice and tissues were homogenized and incubated with a HBSS solution containing 2% BSA (Sigma-Aldrich), 1 mg/ml collagenase D (Roche), and 0.15 mg/ml DNase1, filtered through a 70 μm mesh. Homogenized sections were filtered through 80 μM wire mesh and resuspended in 40% Percoll, prior to density centrifugation (1000 x *g*. 15 min at 20°C with low acceleration and no brake). Cells were acquired on LSRFortessa systems (BD) and analyzed with FlowJo software (Tree Star). For cell sorting, the FacsAria (BD) was used. Antibodies used include: CD11b (clone M1/70; AbD Serotec, Raleigh, NC), CD45 (clone 30F11; AbD Serotec, Raleigh, NC), and MeCP2 (EMD Millipore, Darmstadt, Germany).

## Behavioral visual acuity (optomotor task)

Acuity was measured blind using methods identical to those previously described (*Durand et al., 2012*; *Prusky et al., 2004*).

## Rotarod

Rotarod performance was measured blind using methods similar to those previously described (*Derecki et al., 2012*; *Crawley, 2008*). One day prior to training, mice were acclimated to a non-accelerating rotarod 5 RPM for 5–10 min. The following day, the animals were tested for performance (latency to fall) on an accelerating rotarod over 5 trials, which were subsequently averaged to plot an average latency to fall for each animal.

## Neurological scoring

Neurological scores were recorded blind using methods similar to those previously described (*Derecki et al., 2012*; *Crawley, 2008*). Mice were scored on a scale from 0 to 2, with '0' being no phenotype, and '2' being severe phenotype. For gait, mice were assessed for wide-spread hind limbs and waddling while locomoting. Hind limb clasping was assessed by suspending mice by the tail and assessing clenching of hind limbs across the ventral aspect of the body. Tremor was characterized as a visible involuntary shaking and was scored based on the severity. Appearance was scored based on the presence or lack of grooming and/or hunched posture. The scores were subsequently summed to give a neurological score.

## Statistical analyses

For all statistical analyses, GraphPad Prism 5 software (La Jolla, CA) was used. Analyses used include unpaired Student's t-test, one-way ANOVA, or two-way ANOVA with 95% confidence and appropriate post hoc analyses (indicated in figure legends). All p and N values are indicated in figure legends. All N's represent biological replicates (number of mice used for the study). Sample size was chosen based on our previous work analyzing engulfment and synapse density and work by other groups assessing phenotypic changes in *Mecp2* mutant mice (*Derecki et al., 2012*; *Cronk et al., 2015*; *Schafer et al., 2012*; *Guy et al., 2001*; *Schafer et al., 2014*; *Durand et al., 2012*; *Lioy et al., 2011*; *Chao et al., 2010*; *Guy et al., 2007*; *Patrizi et al., 2016*).

## Acknowledgements

We thank C Chen and M Fagiolini for their guidance and helpful discussions regarding work with *Mecp2* null and mutant animals. We thank G Mandel for the advice regarding detection of Mecp2 in glial cells and M Greenberg for the Mecp2 C-terminal antibody. We also thank W Joyce for his analysis of cell numbers; the imaging core at Boston Children's Hospital including T Hill for his technical support; N Andrews and T Wang for guidance and technical assistance running behavioral experiments at the Boston Children's Hospital Animal Behavior Core; and the Enhanced Neuroimaging core, Harvard Neurodiscovery Center at Harvard Medical School including L Ding and D Tom for their technical support and J Kipnis, J Cronk, and N Derecki for helpful discussions. Work was supported by grants/fellowships from the Nancy Lurie Marks Foundation (DS), NIMH (K99MH102351; DS), NIMH (R00MH102351; DS), John Merck Scholars Program (BS), NINDS (RO1NS07100801-A1; BS), Simons Foundation (BS), International Rett Syndrome Research Foundation (BS), Dana Foundation (BS), NIH (NIH-P30-HD-18655; MRDDRC Imaging Core).

## Additional information

### Funding

| Funder | Grant reference number | Author |
| --- | --- | --- |
| Nancy Lurie Marks Family Foundation | Postdoctoral Fellowship | Dorothy P Schafer |
| National Institute of Mental Health | K99MH102351 | Dorothy P Schafer |

| National Institute of Mental Health | R00MH102351 | Dorothy P Schafer |
| National Institutes of Health | NIH-P30-HD-18655; MRDDRC Imaging Core | Dorothy P Schafer |
| John Merck Fund | Investigator Award | Beth Stevens |
| Simons Foundation | Investigator Award | Beth Stevens |
| International Rett Syndrome Foundation | Investigator Award | Beth Stevens |
| Dana Foundation | Investigator Award | Beth Stevens |
| National Institute of Neurological Disorders and Stroke | RO1NS07100801-A1 | Beth Stevens |

The funders had no role in study design, data collection and interpretation, or the decision to submit the work for publication.

## Author contributions

DPS, Contributed to all aspects of this study including the initial conception and design, data acquisition, and data analysis and interpretation, Wrote and revised the bulk of the manuscript; CTH, Worked closely with Dr. Schafer to design experiments, Performed the majority of data acquisition and data analyses; GG, Acquisition of data, Analysis and interpretation of data, Drafting or revising the article; MH, CG, Acquisition of data, Analysis and interpretation of data; TH, Acquisition of data, Drafting or revising the article; YW, We have added Dr. Yochai Wolf since the original submission. He is a member of Steffen Jung's laboratory and helped us with experiments to validate loss of Mecp2 expression by FACs and DNA., Acquisition of data, Analysis and interpretation of data, Contributed unpublished essential data or reagents; SJ, Drafting or revising the article, Contributed unpublished essential data or reagents; BS, Conception and design, Drafting or revising the article

## Author ORCIDs

Dorothy P Schafer, http://orcid.org/0000-0003-2201-6276

## Ethics

Animal experimentation: This study was performed in strict accordance with the recommendations in the Guide for the Care and Use of Laboratory Animals of the National Institutes of Health. All of the animals were handled according to approved institutional animal care and use committee (IACUC) protocols (# 14-09-2787R) of Boston Children's Hospital. All surgery was performed under isoflurane anesthesia, and every effort was made to minimize suffering.

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
