## [Decision Letter]

Thank you for submitting your article "Microglia contribute to circuit defects in *Mecp2* null mice independent of microglia-specific loss of *Mecp2* expression" for consideration by *eLife*. Your article has been reviewed by two peer reviewers, and the evaluation has been overseen by a Reviewing Editor and a Senior Editor.

The reviewers have discussed the reviews with one another and the Reviewing Editor has drafted this decision to help you prepare a revised submission.

There are two major weaknesses regarding proof of *Mecp2* manipulation and the validity of using homer in the synapse analyses. These should be addressed with new experiments and analyses. Considering how controversial this field is, it is important that the authors demonstrate *Mecp2* knockout in microglia. Once those issues are cleared though, this paper would make a valuable contribution to the field.

Reviewer #1:

MECP2 mutations in humans cause majority of the Rett Syndrome cases, an X-linked neurodevelopmental disorder with severe intellectual and motor phenotypes. Previous studies suggested that the function of this gene in microglia is critically impaired in the disease. Moreover, rescue of some of the disease phenotypes in Rett model mice was observed, following bone marrow transplantation after irradiation. However, other studies did not replicate these findings, making it unclear whether microglial *Mecp2* function is involved in Rett syndrome pathogenesis.

This manuscript by Dr. Schafer and colleagues investigate the function of *Mecp2* gene specifically in microglial synaptic pruning using the mouse retinogeniculate system.

The manuscript starts with the demonstration of a previously uncharacterized early adult pruning event that occurs around P40 in the wildtype mouse geniculate. They find that synaptic pruning, measured as the engulfment of labeled RGC terminals, is unaffected in P5-P50 *Mecp2* KO mice compared to WT. But in *Mecp2*-KO mice (that are older than 56 days) there is enhanced axonal terminal engulfment and lysosomal content in microglia. They then use global null and conditional loss- or gain-of-function alleles of *Mecp2* in combination with CX3CR1-CreER to specifically silence or rescue the expression of *Mecp2* in microglia. They find that manipulations of *Mecp2* expression in microglial have little effect on excessive engulfment, structural and functional visual circuit defects, or other general phenotypes associated with *Mecp2*-function. Based on these results the authors conclude that microglial contribution to Rett syndrome is likely to be secondary after the loss of *Mecp2* function in other CNS cell types.

The article is well-written, and the reported experimental data provide compelling evidence in support of the authors' conclusions. Use of multiple genetic approaches in combination with rigorous analysis of microglial phenotypes is impressive. However, the manuscript currently does not present any evidence for effective *Mecp2* gene manipulation using CX3CR1-CreER. Without such evidence we cannot exclude the possibility that the observed minimal phenotypes are due to ineffective manipulation of *Mecp2* expression in microglia by CX3CR1-CreER. Moreover, there are several other concerns with regards to how the synapses were quantified, which are detailed below. The authors should provide new data and analyses to address these concerns.

1) Lack of evidence showing effective modification of *Mecp2* gene in microglia by CX3CR1-CreER. CX3CR1-CreER transgenic mouse line was thoroughly characterized and confirmed as a good tool to silence gene expression in microglia. However, (as with any Cre line used in a conditional gene manipulation paradigm) the authors should demonstrate that using their tamoxifen injection paradigm, in their brain region of interest, the microglial expression of *Mecp2* is altered. Without such assurance it is hard to exclude that ineffective *Mecp2* manipulation in microglia in general or the geniculate microglia in particular leads to the observed weak phenotypes. Considering how controversial this particular field is, the authors should provide this crucial control so their results are fully supported. Effectiveness of this manipulation in multiple animals should be demonstrated to be able to exclude the possibility of variability between individual mice.

2) Quantification of synapse number in the geniculate. The authors use an imaging-based technique to estimate geniculate synapses. This technique, the quantification of co-localized pre and postsynaptic protein signals, is appropriate; however, the provided images show that homer staining is very dense and potentially mostly not synaptic. At the resolution and magnification of the provided images it is hard to conclude whether VGlut2 and homer are truly co-localizing or is it just pure chance. Also the materials and method section describing how this quantification was done is not detailed. Where in geniculate did the pictures originate? It is important that the images were acquired from the same area in different animals so they could be comparable. How were the images acquired? Was the analysis performed on single optical sections or maximum projections? Also VGlut1 synapses were quantified as well, but no representative images are provided, so it is hard to evaluate the quality of staining on which that analysis was based.

Reviewer #2:

In this paper Schafer and colleagues investigate the role of microglia in the Rett syndrome (*Mecp2* null mouse). They first focus on the pruning role of microglia and find that this is normal at the onset of the disease but it increases at later stages. They also demonstrate that gain and -most importantly- loss of the *Mecp2* gene in microglia has little effect on engulfment and on characteristic Rett phenotypes. Thus, the main conclusion here is that microglia cannot be considered major contributors in Rett pathology.

This is clearly a "non-discovery" but it is an important one because it resolves conflicting reports on this matter. The controversy is exemplified but the clashing titles of the two major publications on this topic: "Wild-type microglia arrest pathology in a mouse model of Rett syndrome" (Derecki et al. Nature 2015) and "Wild-type microglia do not reverse pathology in mouse models of Rett syndrome" (Wang et al. Nature 2015). Thus, I believe the work of Schafer and colleagues to be of value and to deserve publication in a rigorous and respected journal.

There are few points, however, that need to be clarified and better demonstrated.

1) The authors show that at later stages *Mecp2* microglia have more phagosomes containing pre-synaptic inputs. This increase can be explained by active synaptic remodeling but also by engulfment of neurons within the *Mecp2* brain. They exclude the latter by comparing the number of neurons and nuclei in wildtype and *Mecp2* null. Considering the large number of cells in the brain this cannot be considered a sensitive method and needs to be backed up by a direct visualization and quantification of neuronal (non-synaptic) material in microglia. If microglia are involved in active synaptic remodeling they should not show increased -generic- neuronal markers in their phagosomes. This point is critical also considering that in *Mecp2* null microglia there 3 times more lysosomes (Figure 2). Considering that only a smaller fraction contains synaptic material (compare graph in Figure 2 with Figure 2) the authors should exclude that these vesicles contain non-synaptic material resulting from the engulfment of entire neurons.

2) Throughout this paper the authors write that microglia at later stages in the *Mecp2* null mouse remove weaken synapses. That microglia can remove weaken synapses was shown by Schafer and colleagues in their 2012 Neuron paper. Here, however, they give no evidence that in *Mecp2* microglia remove weaker synapses. They should remove these statements or provide evidence for this.

3).Please, note that in. Figure 1—figure supplement 1 the postsynaptic protein Homer1 is visible at P30 (iii) but not at P60 (iii) nevertheless the staining is quantified in E.

4) In Figure 1—figure supplement 2 it is written that engulfment of presynaptic inputs is significantly increased over old ages. Judging from the graph the opposite is true.

Reviewer #2 (Additional data files and statistical comments):

The authors have stained *Mecp2* and wild-type with an anti-activated-Caspase 3 antibody. Please, include these data.

---

## [Author Response]

There are two major weaknesses regarding proof of Mecp2 manipulation and the validity of using homer in the synapse analyses. These should be addressed with new experiments and analyses. Considering how controversial this field is, it is important that the authors demonstrate Mecp2 knockout in microglia. Once those issues are cleared though, this paper would make a valuable contribution to the field.

We have added new data to existing figures, added an additional 3 supplemental figures (Figure 2—figure supplement 2, Figure 4—figure supplement 1, and Figure 5—figure supplement 1), and made substantial revisions to the text to address all reviewer concerns. Please see detailed responses to each concern.

Reviewer #1:

*1) Lack of evidence showing effective modification of Mecp2 gene in microglia by CX3CR1-CreER. CX3CR1-CreER transgenic mouse line was thoroughly characterized and confirmed as a good tool to silence gene expression in microglia. However, (as with any Cre line used in a conditional gene manipulation paradigm) the authors should demonstrate that using their tamoxifen injection paradigm, in their brain region of interest, the microglial expression of Mecp2 is altered. Without such assurance it is hard to exclude that ineffective Mecp2 manipulation in microglia in general or the geniculate microglia in particular leads to the observed weak phenotypes. Considering how controversial this particular field is, the authors should provide this crucial control so their results are fully supported. Effectiveness of this manipulation in multiple animals should be demonstrated to be able to exclude the possibility of variability between individual mice.*

We have spent significant time and resources addressing this particular concern. One major issue was that attaining enough RNA or protein from microglia isolated from the LGN for qPCR or western blot was not technically feasible and/or required a large amount of animals from lines of mice that do not reproduce well. As a result, we attempted immunohistochemistry on our existing tissue (from engulfment and synapse analysis experiments) to detect *Mecp2* protein in LGN microglia. For several weeks we tried many different antibodies and staining conditions and were never able to detect *Mecp2* protein even in wild-type microglia (see Figure 6 for a subset of attempts). This is likely due to very low expression of *Mecp2* in microglia (Zhang et al. J. Neurosci 2014). However, we finally achieved success using a non-commercially available rabbit *Mecp2* antibody developed by and obtained from Dr. Michael Greenberg. Using this antibody directed against the C-terminus of *Mecp2* combined with antigen retrieval and HRP amplification (Ballas et al. Nat Neurosci2009), we were able to detect low levels of *Mecp2* protein in 40-50% of wild-type LGN microglia and *Mecp2* expression was either lost with recombination in *Cx3CR1_CreER/WT_;Mecp2_fl/y_*+TAM or *Cx3CR1_CreER/WT_;Mecp2_LSL/y_*+Oil mice or gained in *Cx3CR1_CreER/WT_;Mecp2_LSL/y_*+TAM mice (see new Figures: Figure 4—figure supplement 1 and Figure 5—figure supplement 1). In addition, we performed other experiments to show sufficient recombination including analysis of efficiency and specificity of tdTomato expression following tamoxifen injection in *Cx3CR1_CreER/WT_*;Rosa26-tdTomato mice (see new Figure 4—figure supplement 1). We also performed PCR on genomic DNA and FACs analysis to assess protein expression in microglia isolated from the whole brain of *Cx3CR1_CreER/WT_;Mecp2_fl/y_*+TAM mice (see new Figure 4—figure supplement 1).

Author response image 1.Attempts to immunostain for *Mecp2* protein in WT LGN microglia.To validate loss or gain of *Mecp2* expression, we first required an antibody to recognize *Mecp2* protein in WT microglia. We made numerous attempts before we were able to detect adequate levels for analysis (see Figure 4—figure supplement 1 and Figure 5—figure supplement 1). Failed attempts are in LGN microglia are shown. Merge channels are on top row and the *Mecp2* channel is in the bottom row. Arrows denote microglia nuclei and lack of *Mecp2* immunostaining in bottom panel. (**A**) Immunostaining with a Millipore chicken anti- *Mecp2* antibody with antigen retrieval (red in Ai, Aii) and anti-Iba-1 to label microglia. (B-C) Immunostaining with a C-terminal anti-*Mecp2* antibody (gift from M. Greenberg from Harvard Medical School; red in Bi and Ci, Bii and Cii) and anti-Cd11b to label microglia without (**B**) and with (**C**) amplification using HRP. We were successful detecting *Mecp2* in microglia with this antibody following antigen retrieval and HRP amplification (see Figure 4—figure supplement 1 and Figure 5—figure supplement 1). (**D**) Immunostaining with a Millipore rabbit anti-Mecp2 antibody with HRP amplification (red in Di, Dii) and anti-Cd11b to label microglia. All Scale bars =10 µm.**DOI:**
http://dx.doi.org/10.7554/eLife.15224.015

2) Quantification of synapse number in the geniculate. The authors use an imaging-based technique to estimate geniculate synapses. This technique, the quantification of co-localized pre and postsynaptic protein signals, is appropriate; however, the provided images show that homer staining is very dense and potentially mostly not synaptic. At the resolution and magnification of the provided images it is hard to conclude whether VGlut2 and homer are truly co-localizing or is it just pure chance. Also the materials and method section describing how this quantification was done is not detailed. Where in geniculate did the pictures originate? It is important that the images were acquired from the same area in different animals so they could be comparable. How were the images acquired? Was the analysis performed on single optical sections or maximum projections? Also VGlut1 synapses were quantified as well, but no representative images are provided, so it is hard to evaluate the quality of staining on which that analysis was based.

The main conclusion of this part of the paper is that we see changes in presynaptic inputs and thus a decrease in structural synapses (juxtaposed pre and postsynaptic elements). There is no significant change in the postsynaptic (or even extrasynaptic) Homer immunoreactivity. We have updated the manuscript to better clarify this point and included quantification of Homer immunoreactivity (see updated Figure 3). In addition, the representative images in the original manuscript (Figure 3) were maximum intensity projections as we felt this most fully demonstrated the loss of presynaptic inputs (the main conclusion). However, this was at the expense of postsynaptic staining appearing relatively non-specific and dense. Given that the analysis was actually performed on single confocal planes, we modified this figure to include single planes which better show co- localization (see updated Figure 3).

The reviewer also raises questions regarding the synaptic localization of Homer. We originally chose Homer given that, out of all postsynaptic markers, it gave us the most consistent and strong signal to noise. Interestingly, retinogeniculate synapses represent <10% of presynaptic inputs in the LGN (Bickford et al. J Comp Neurol2009). We have noted this in the revised manuscript. Thus, while it is appreciated that some Homer immunoreactivity could be extrasynaptic, our data are consistent with a large proportion of Homer immunoreactivity as postsynaptic and colocalized with other non- RGC (non-VGlut2 positive) terminals. This is demonstrated by increased co-localization of Homer/VGlut1 versus Homer/ VGlut2 (see updated Figure 3—figure supplement 1 and Figure 7). We have also included Figure 8 demonstrating that VGlut2 terminals also co- localize with another postsynaptic marker, anti-PSD-95, to a similar degree as Homer. Furthermore, to address whether colocalization occurs purely by chance, we rotated the VGlut2 channel 90 degrees and performed co-localization analysis. This analysis demonstrates that there is higher cumulative probability of colocalization in correctly oriented VGlut2/Homer or VGlut2/PSD-95 images as compared to those in which the VGlut2 channel was rotated (see Figure 8).

Author response image 2.VGlut1-containing synapses are more abundant than VGlut2- containing synapses within the dLGN.Quantification of the density of VGlut1 or VGlut2 immunoreactive puncta colocalized with Homer. Data were collected from 4-5 P60 wild-type mice. Error bars represent ± SEM.**DOI:**
http://dx.doi.org/10.7554/eLife.15224.016

Author response image 3.Postsynaptic proteins colocalize with VGlut2 more frequently over chance.(**A**) Single plane of a confocal z-stack in the adult wild-type dLGN immunostained with anti-VGlut2 (white), anti-Homer (green, Aii), and anti-PSD-95 (red, Aiii). Scale bar=10 µm. (**B–C**) Cumulative probability of co-localized Homer (**B**) or PSD- 95 (**C**) with VGlut2 before (solid line) or after 90 degree rotation (dashed line) of the VGlut2 channel. Colocalization of both postsynaptic proteins is more probable over chance (90 degree rotation). Data were collected and averaged across 3 adult wild- type animals (2 single plane images from the same region were collected and analyzed for each animal). D.Result of colocalization analysis between VGlut2 and Homer (Di) and VGlut2 and PSD-95 (Dii). Result of colocalization upon 90 degree rotation of the VGlut2 channel.**DOI:**
http://dx.doi.org/10.7554/eLife.15224.017

As requested by the reviewer, we have also included representative images of VGlut1/Homer staining (see updated Figure 3—figure supplement 1) and we updated the Materials and methods section to more extensively describe the data collection. Briefly, for each animal, 3 confocal z stacks were taken in the same regions of the dLGN. Then, 2 planes out of each z-stack were chosen blindly based on the most abundant DAPI staining for analysis. As a result, a total of 6 images were analyzed in nearly identical locations were analyzed across animals.

*Reviewer #2:*

*There are few points, however, that need to be clarified and better demonstrated.*

1) The authors show that at later stages Mecp2 microglia have more phagosomes containing pre-synaptic inputs. This increase can be explained by active synaptic remodeling but also by engulfment of neurons within the Mecp2 brain. They exclude the latter by comparing the number of neurons and nuclei in wildtype and Mecp2 null. Considering the large number of cells in the brain this cannot be considered a sensitive method and needs to be backed up by a direct visualization and quantification of neuronal (non-synaptic) material in microglia. If microglia are involved in active synaptic remodeling they should not show increased -generic- neuronal markers in their phagosomes. This point is critical also considering that in Mecp2 null microglia there 3 times more lysosomes (Figure 2). Considering that only a smaller fraction contains synaptic material (compare graph in Figure 2 with Figure 2) the authors should exclude that these vesicles contain non-synaptic material resulting from the engulfment of entire neurons.

We appreciate the reviewers comment and have done more analysis of engulfment of non- synaptic material including NeuN immunoreactive neuron cell somas and Map2-immunoreactive somato-dendritic domains within the LGN (see updated Figure 2—figure supplement 2). We do not observe an increase in engulfment of these neuronal compartments as compared to presynaptic input engulfment. This is consistent with our data and previously published work demonstrating no neuronal cell death in the mouse models of Rett Syndrome or in human patients (reviewed in Chahrour and Zoghbi Neuron 2007; Banerjee et al. Front Psychiatry 2012). This reviewer does raise an interesting point as to what exactly, if anything, occupies these remaining lysosomes. While outside the scope of the current manuscript, this would be an interesting future direction.

2) Throughout this paper the authors write that microglia at later stages in the Mecp2 null mouse remove weaken synapses. That microglia can remove weaken synapses was shown by Schafer and colleagues in their 2012 Neuron paper. Here, however, they give no evidence that in Mecp2 microglia remove weaker synapses. They should remove these statements or provide evidence for this.

We suggest that microglia engulf weakened synapses. This is based on previous electrophysiology demonstrating an overall weakening of retinogeniculate synapses by P30 (Noutel et al. Neuron 2011), a timepoint preceding any significant signs of engulfment. While we still suggest this as a possibility in the manuscript, we have toned down the language and removed this from the overall conclusions.

3) Please, note that in. Figure 1—figure supplement 1 the postsynaptic protein Homer1 is visible at P30 (iii) but not at P60 (iii) nevertheless the staining is quantified in E.

These data do appear in our pdf version of the manuscript, which was downloaded for review at *eLife*. It should be noted that the Homer staining pattern does differ across ages (soma and small puncta staining at P30 versus only puncta staining at P60). This point was raised by Reviewer 1 and it may have some biological significance and differences in extrasynaptic (somatic) versus synaptic (small puncta) homer localization.

4) In Figure 1—figure supplement 2 it is written that engulfment of presynaptic inputs is significantly increased over old ages. Judging from the graph the opposite is true.

We apologize. This was poorly written. We modified the figure legend to better reflect that, as the reviewer correctly points out, engulfment is higher at P5 and P40 versus P60.

*Reviewer #2 (Additional data files and statistical comments):*

*The authors have stained Mecp2 and wild-type with an anti-activated-Caspase 3 antibody. Please, include these data.*

We have included these data in the revised manuscript (see updated Figure 2—figure supplement 1).